# Intestinal Epithelial Cells Adapt to Chronic Inflammation through Partial Genetic Reprogramming

**DOI:** 10.3390/cancers15030973

**Published:** 2023-02-03

**Authors:** Guillaume Collin, Jean-Philippe Foy, Nicolas Aznar, Nicolas Rama, Anne Wierinckx, Pierre Saintigny, Alain Puisieux, Stéphane Ansieau

**Affiliations:** 1Centre de Recherche en Cancérologie de Lyon, INSERM U1052, CNRS UMR 5286, Centre Léon Bérard, Université Lyon1, 69008 Lyon, France; 2ProfileXpert, 69008 Lyon, France; 3Department of Medical Oncology, Centre Léon Bérard, 69008 Lyon, France

**Keywords:** adaptation to chronic-inflammation-associated oxidative stress, intestinal cell reprogramming, ZEB transcription factors, inflammatory bowel disease progression

## Abstract

**Simple Summary:**

Chronic inflammation, as observed in Crohn’s disease and ulcerative colitis patients, damages the intestinal mucosa. We reasoned that if a subset of intestinal epithelial cells adapt to this inflammatory stress to survive, this adaptation could contribute to their malignant transformation. We show that human colonic epithelial cells escape chronic inflammation in vitro through a partial genetic reprogramming. By questioning data bases, we confirm that this reprogramming takes place in the inflamed mucosae of patients with Crohn’s disease and ulcerative colitis, and that it is induced in vivo during the early stages of murine intestinal carcinogenesis. These data contribute to understanding the pathology and underline how orchestrators of cellular adaptation might contribute to intestinal homeostasis and, potentially, tumor initiation.

**Abstract:**

Reactive oxygen species (ROS) are considered to be the main drivers of inflammatory bowel disease. We investigated whether this permanent insult compels intestinal stem cells to develop strategies to dampen the deleterious effects of ROS. As an adverse effect, this adaptation process may increase their tolerance to oncogenic insults and facilitate their neoplastic transformation. We submitted immortalized human colonic epithelial cells to either a mimic of chronic inflammation or to a chemical peroxide, analyzed how they adapted to stress, and addressed the biological relevance of these observations in databases. We demonstrated that cells adapt to chronic-inflammation-associated oxidative stress in vitro through a partial genetic reprogramming. Through a gene set enrichment analysis, we showed that this program is recurrently active in the intestinal mucosae of Crohn’s and ulcerative colitis disease patients and evolves alongside disease progression. Based on a previously reported characterization of intestinal stem and precursor cells using tracing experiments, we lastly confirmed the activation of the program in intestinal precursor cells during murine colorectal cancer development. This adaptive process is thus likely to play a role in the progression of Crohn’s and ulcerative disease, and potentially in the initiation of colorectal cancer.

## 1. Introduction

Chronic inflammation of the intestinal epithelium, observed in patients with inflammatory bowel diseases (IBD) such as ulcerative colitis (UC) and Crohn’s disease (CD), damages cell integrity, mainly through the accumulation of reactive oxygen species (ROS) and nitric oxygen species (NOS), which are produced by infiltrating leukocytes, activated macrophages, and neutrophils [1]. Experimental studies unquestionably demonstrate that oxidative stress is a determinant factor in the pathogenesis of IBD and the associated carcinogenesis [2,3]. This stress causes lipid and protein oxidation, DNA damage, telomere attrition, and, consequently, cellular senescence [4,5]. Telomere attrition induces the fusion of chromosomal ends, resulting in chromatin bridge breakage and fusion and chromosomal instability, which is associated with tumor progression. In IBD patients, telomere attrition is thus considered to be a consequence of inflammation [6]. Conversely, a set of recent studies supports that telomere shortening promotes inflammation. Indeed, *mTert* deletion in murine intestinal epithelial cells was shown to lead to the activation of an ATM-YAP-pro-IL-18 pathway, a subsequent induction of IFNϒ secretion by resident T cells, and consequently to tissue inflammation [7]. Telomere disruption and pro-IL-18 production was confirmed in IBD-patient-derived organoids, and telomere reactivation, suppression of DNA damage, or YAP inactivation was shown to be sufficient to suppress IL-18 production [8].

*TP53* mutations are recurrent in both sporadic and IBD-associated colorectal cancer (CRC). While considered as a late event in sporadic CRC [9,10], *TP53* mutations occur very early in patients with IBD-associated CRC, before neoplastic deletion becomes detectable [11,12], and have been associated with NOS2 activity [13]. These mutations are multiclonal in inflamed mucosae and low-grade dysplasia and monoclonal in high-grade dysplasia [14]. The *TP53* mutation may thus provide a selective advantage at a later stage, likely by facilitating an escape from replicative senescence. As demonstrated in engineered mouse models, escape relies on the ability of mutated p53 isoforms to turn cancer-inhibitory properties of the senescent cell secretome into cancer-promoting properties in intestinal epithelial cells [15]. Moreover, a mutant p53 was reported to increase IBD-associated CRC incidence by promoting aneuploidy [12] and by sustaining NF-κB activation, thereby exacerbating inflammation [16].

In addition to the pro-tumoral properties of ROS, we herein investigated whether a subset of intestinal epithelial stem cells (or inflammation-driven dedifferentiated cells [17]) could acquire a resistance to ROS during disease progression, thus becoming more tolerant to long-term oncogenic insults and more permissive to neoplastic transformation. In line with this hypothesis, SIRT6-driven metabolic adaptation was recently shown to suppress ROS accumulation in a subset of stem cells and to enhance their tumorigenic potential [18]. In an attempt to evaluate this possibility, we set up an in vitro model subjecting human colonic epithelial cells to either oxidative stress (by a chemical peroxide) or a mimic of chronic inflammation. We identified a novel genetic program (hereafter referred to as OSAP for Oxidative Stress Adaptation Program) which provides cells with a survival advantage, evaluated its potential induction in stem/early precursor cells, and investigated its biological significance in IBD patients and colorectal tumors using a single-sample gene set enrichment analysis (ssGSEA).

## 2. Materials and Methods

### 2.1. Constructs and Cell Lines

The *Zeb1* and *Zeb2* pbabe-Puro constructs were previously described [19]. Human primary colonic epithelial cells (HCEC) were purchased from Applied Biological Materials (abm, Richmond, Canada) and immortalized through transduction of *hTERT* (HCEC-hTERT). THP-1 and 293T were purchased from the ATCC. Cell lines were cultured according to the supplier’s recommendations, and experiments were performed at early passages. Human *APOE* (variant ε3) and *QPRT* pbabe Puro retroviral constructs were derived from the pLenti-GIII-hAPOE-GFP-2A-Puro constructs (abm) and QPRT pCMV6 (Origene, Herford, Germany), respectively.

### 2.2. Retroviral and Lentiviral Infections

HCEC-hTERT *Zeb1*, *Zeb2*, *QPRT,* or *APOE* cell lines were generated by retroviral infections as previously described [20]. Briefly, cells were “murinized” by expressing the ecotropic retrovirus receptor before being infected with the retroviral expression constructs. The second infection was conducted after 48 h. Selection was initiated at 24 h after the second infection using puromycin (1.5 µg/mL). 

### 2.3. TP53 Status

The TP53 status in parental HCEC-hTERT cells and derivatives was first defined by sequencing (Sanger technic). An assessment of the functionality of the p53 pathway was achieved by submitting parental HCEC-hTERT and their derived Esc-Inf cells to increasing doses of 5-FU (Accord) for 24 h (0, 50, 200, 350, 500, and 1000 µM).

### 2.4. Oxidative Stress Induction in HCEC-hTERT Cells

THP-1 monocyte differentiation (5 × 10^7^ cells) was induced by adding 0.162 µM of PMA (Sigma Aldrich Chimie, Saint-Quentin-Fallavier, France) for one day, and the activation of the resulting macrophages was then induced by treating with LPS (LPS-EB ultrapure from *E. coli* 0111:B4 strain, InvivoGen, Toulouse, France) for an additional day. The cell supernatant was collected, filtered at 0.45 µM, diluted 1/2 in HCEC culture medium, and incubated on HCEC-hTERT cells (5 × 10^5^ in a 6-well dish). The activated macrophage supernatant was renewed every two days. Similarly, treatment of HCEC-hTERT cells with tert-Butyl-hydroperoxide (TBHP, 30 µM, Sigma) was renewed every two days. The HCEC-hTERT cells were stained with crystal violet after 10 days of treatment, and the stained surfaces were quantified using the Image J software. SA-β-galactosidase assays were performed as described previously [21]. 

### 2.5. Gene Expression Analysis by qRT-PCR

RNA preparation and reverse transcription were performed as previously described [20]. Real-time PCR intron-spanning primers were designed with the primer3 software. The *HPRT1* housekeeping gene was used for normalization. The list of primers used is accessible in the Appendix A.

### 2.6. Gene Expression Profiling of Cell Lines

The microarray processing and data analysis of the HCEC-hTERT parental and derived cells were performed at the ProfileXpert core facility (Lyon, France). Gene expression profiles were analyzed with a whole human genome microarray containing 47,231 probes (Human HT-12 v4 Expression BeadChip; Illumina Inc., San Diego, CA, USA). Total RNA (500 ng) was amplified and biotin-labeled with the Illumina TotalPrepTM RNA Amplification Kit (Ambion Inc., Austin, USA). Hybridization was performed with 750 ng of biotin-labeled cRNA on each BeadChip. The standard Illumina scanning protocol was used to scan the arrays with the iScan (Illumina Inc., U.S.A.). Data were normalized by quantile normalization using the Genome Studio Software 2010 (Illumina Inc.). The complete set of raw and normalized files is available at the GEO database under accession number GSE70170.

### 2.7. Immunoblot Analysis

Cells were lysed in a 100 mM NaCl, Tris 50 mM pH8, NP40 1%, Glycerol 50% extraction buffer supplemented with a complete protease inhibitor cocktail (Roche, Meylan, France). After sonication, extracts were clarified by centrifugation, quantified by Bradford staining (Bio-Rad, Marnes-la-Coquette, France), denatured by heating (95 °C for 3 min) and separated by SDS-PAGE. Protein expression was examined using either the monoclonal anti-p21^CIP1/WAF1^ clone SX118 (Agilent, Les Ulis, France), anti-QPRT (ab57125, Abcam), anti-apolipoprotein E clone E6D7 (Abcam), anti-p53 DO-7 (Agilent), anti-α-tubulin clone DM1A (Sigma), or the anti-β-actin clone AC-15 (Sigma) antibody, or the polyclonal anti-ZEB1 H102 (Santa Cruz Biotechnology, Heildelberg, Germany), anti-ZEB2 [22], anti-phospho-histone H3 (Ser10)-R (Santa Cruz Biotechnology), anti-phospho p53 (Ser15) (Cell signaling), anti-acetyl p53 (Lys373, Lys 382) (Millipore SAS, Molsheim, France), and the NF-κB p65 (C20) (Santa Cruz Biotechnology) antibody and horseradish-peroxidase-conjugated secondary antibodies (Dako). Western blots were normalized using the Image J software.

### 2.8. In silico Analysis for Validation of the Experimental OSAP

An examination of the relevance of the OSAP in independent datasets from Gene Expression Omnibus (GEO), GSE39582 [23], GSE37283 [24], GSE36807 [25], GSE16879 [26], GSE36133 [27], GSE 46200 [28], and GSE37929 [29] was performed using the Array Studio software (Qiagen France SAS, Les Ulis, France). The methodologies employed to process and normalize data and to perform the Hierarchical Clustering Analysis, Principal Component Analysis (PCA), and ssGSEA projections are detailed in the Appendix A.

## 3. Results

### 3.1. Human Colonic Epithelial Cells Exposed to Chronic Inflammation Develop a Resistance Mechanism

Human primary colonic epithelial cells (HCEC, abm) were sequentially immortalized through *hTERT* transduction (HCEC-hTERT cells) and cultured in the presence of activated macrophage supernatants (AMS) to mimic chronic inflammation or a chemical peroxide (tert-Butyl-hydroperoxide, TBHP) to specifically address the effects of ROS (Figure 1A). Conditions were set up so that cells were submitted to sublethal doses, experimental conditions commonly used to enforce cell adaptation and unveil escape mechanisms. In both experimental settings, after four to five days, cells stopped proliferating (a reduction of phospho-Serine 10 histone H3, pS10-H3) and were committed to a senescence program, as revealed by the accumulation of acetylated p53 (Ac-p53) and its target p21^CIP1^ (p21) and the detection of SA-β-galactosidase activity (Figure 1B). Despite the fact that telomere length is regulated by ectopic telomerase expression, our cellular model recapitulated some in vivo observations [30,31]. Cells maintained their senescent phenotype for several weeks under stress. Strikingly, after a month in both experimental settings, cells uniformly and concomitantly resumed proliferation, supporting an adaptive process rather than the selection of preexisting clones. The emerging cells were hereafter named Esc-Inf for “Escape from chronic inflammation-induced senescence” or Esc-TBHP for “Escape from TBHP-induced senescence” (Figure 1A).

### 3.2. Resistance to Oxidative Stress Is p53-Independent

We next sought to identify the mechanisms at the basis of this escape. The p53 transcription factor is a main regulator of cellular senescence and its loss of function is frequently associated with an escape from this safeguard program mechanism in the early stages of tumor development [32,33]. In the context of IBD, *TP53* monoclonal mutations observed in high-grade dysplasia of patients with ulcerative colitis were suggested to facilitate escape from oxidative-stress-induced senescence [34]. Hence, we evaluated whether the loss of function of p53 contributed to this proliferation/survival advantage by sequencing *TP53* mutation hot-spots in Esc-Inf (Esc-Inf A and B from two independent experiments) and parental HCEC-hTERT cells. The wild-type status was confirmed in both the parental cell line and its derivatives (Figure 2A). Furthermore, the activity of the p53 pathway was similar in these cell lines, as assessed by p53-phosphorylation (pS15-p53) and the subsequent accumulation of p21^CIP1^ (p21) in response to genotoxic stress (Figure 2B). Lastly, the transduction of parental HCEC-hTERT cells with p53 variants that constitute the mutational hot-spots observed in ulcerative colitis dysplastic and malignant lesions [35,36] failed to protect cells from chronic-inflammation-induced senescence (Figure 2C). An adaptation to an oxidative stress condition thus did not rely on the alteration of p53 activity in these experimental conditions. 

### 3.3. Adaptation to Oxidative Stress Relies on Partial Cell Reprogramming

Having excluded that the mechanism of escape relies on a loss of p53 function, we next assessed the possibility that the emergence of resistant cells was dependent on the induction of embryonic factors of the TWIST, SNAIL, and ZEB families, known to facilitate escape from replicative and oncogenic senescence and cell adaptation to stress [37,38,39,40,41,42].

Escapes from chronic-inflammation- or TBHP-induced senescence were invariably associated with *ZEB1* or *ZEB2* induction, respectively, while *SNAI1* and *TWIST2* RNA levels rose marginally (*SNAI2*, *SNAI3,* and *TWIST1* expression were below the detection limit) (Figure 3A). A protein assessment by Western blot analysis confirmed the detection of ZEB1 in Esc-Inf cells, though the protein level remained low compared to EMT-committed Hs578T breast cancer cells (Figure 3A). Of note, we failed to detect ZEB2 in Esc-TBHP cells, likely due to the combination of a low protein level and the low affinity of the antibody. The selection of either *ZEB1* or *ZEB2* in the two experimental settings strongly argues in favor of a stress-induced cell reprogramming rather than the selection of preexisting cell subpopulations. We then expected that an ectopic expression of *ZEB* genes would provide cells with the observed survival/proliferation advantage. To address this hypothesis, HCEC-hTERT cells were transduced with murine *Zeb1* or *Zeb2* retroviral-expressing constructs (HCEC-*Zeb1* cells and HCEC-*Zeb2* cells, respectively) with similar ranges of protein levels to those observed in spontaneously emerging cells (ZEB1 was barely detectable by Western blot and ZEB2 was below the detection limit) (Appendix A). We confirmed that these transcription factors provided the HCEC-hTERT cells with a similar survival/proliferation advantage in both stress conditions, as assessed in a crystal violet coloration assay (Figure 3B). 

To explore the underlying mechanism, we next compared the gene expression profiles of cells that either escaped from chronic-inflammation-induced (Esc-Inf A and B from two independent experiments) or TBHP-induced (Esc-TBHP A and B from two independent experiments) senescence (maintained over a week in the absence of stress to stably identify up- and down-regulated genes), and those of the HCEC-*Zeb1* and HCEC-*Zeb2* cell lines to their HCEC-hTERT parental counterparts (dataset GSE70170) (Figure 4A). A core of 27 up-regulated and 32 down-regulated genes (using a cut-off of 1.5) was commonly modulated in these cell lines (Figure 4B). This genetic program was hereafter referred to as the Oxidative Stress Adaptation Program (OSAP). A gene ingenuity pathway analysis unveiled that the down-regulated genes were enriched in NF-κB targets (*p* = 4.4·10^−5^, Appendix A). Indeed, RelA was found to be abnormally activated in the parental HCEC-hTERT cells and down-regulated in the Esc cells, as revealed by its subcellular localization (Appendix A). The repression of NF-κB-target genes in the Esc cells is most likely a consequence of *APOE* induction, which is an inhibitor of RelA [43,44,45]. No specific pathway was enriched from the OSAP-associated, up-regulated genes (*p* < 10^−2^, Appendix A), including the epithelial-to-mesenchymal transition, a genetic reprogramming also orchestrated by ZEB proteins [46].

To further strengthen the role of the OSAP in providing cells with a survival advantage under oxidative stress conditions, the two most highly up-regulated genes, *QPRT* and *APOE,* were first confirmed to be induced in the Esc-Inf cells and Esc-TBHP cells compared to the parental cell line at both transcript and protein levels (Figure 4C). They were then tested for their ability to protect HCEC-hTERT cells from chronic-inflammation- and TBHP-induced senescence. As is shown in Appendix A, the ectopic expression of either QPRT or APOE significantly sustained the HCEC-hTERT cell proliferation under oxidative stress, suggesting that several OSAP-associated proteins likely contribute to the ZEB-driven survival advantage.

### 3.4. OSAP Is Activated in Ulcerative Colitis and Crohn’s Disease Mucosa

Having shown that colonic epithelial cells adapted to chronic-inflammation-induced oxidative stress in vitro through a partial genetic reprogramming, we then sought to confirm the relevance of these observations in human patients. To this end, we assessed the score of the OSAP-associated genetic signature in inflammatory bowel disease patients by ssGSEA.

We initially focused on a series of patients with an established diagnosis of ulcerative colitis (UC) or Crohn’s disease (CD) and with a disease duration > 7 years with no macroscopic sign of active inflammation (datasets GSE37283 and GSE36807) [24,25]. Unsupervised hierarchical clustering of UC and CD cases using the in vitro OSAP signature confirmed a differential expression of genes between samples (Figure 5A). Using a Wilcoxon test, the comparison of the UC mucosa versus healthy controls (dataset GSE36807) or the UC mucosa with neoplasms versus healthy controls (dataset GSE37283) highlighted a co-segregation of eleven induced and four repressed genes in both series (Figure 5B). This refined signature was used as a readout of the OSAP induction in human samples with cellular heterogeneity. Of note, most of the in vitro down-regulated genes associated with the OSAP signature were excluded from the refined signature. These genes are mainly targets of NF-κB and are activated in UC and CD patients owing to chronic inflammation. 

We next used this refined OSAP signature to address its biological relevance. Gene expression profiles of CD and UC patients of the dataset GSE36807 previously demonstrated that healthy samples clustered together, while variations in the gene expression patterns between the CD and UC patients were more complex, highlighting a heterogeneity within IBD samples [25]. As is shown in Figure 6A,B, an unsupervised, hierarchical clustering of cases using the refined OSAP signature segregated a subgroup of CD and UC patients, suggesting that the OSAP was commonly activated in both pathologies. 

We then investigated whether the OSAP activation parallels disease progression by scoring the refined OSAP signature in a dataset (GSE37283) previously used to unveil differential gene expression in the remote, quiescent, non-dysplastic mucosa of patients harboring neoplastic lesions compared to the mucosa of UC patients without dysplasia and normal controls [24]. Gene ontology revealed a differential activation of innate immune response pathways and Toll-like receptors, confirming that neoplastic lesions in UC likely resulted from the activation of tumor-promoting pathways secondary to longstanding infection [24]. An unsupervised, hierarchical clustering of the OSAP signature-associated genes discriminated non-dysplastic mucosa from UC patients that harbored remote neoplastic lesions from those of UC patients without dysplasia. Furthermore, patients with UC-associated neoplasms displayed significantly higher OSAP signature scores (Mann–Whitney Test, *p* = 0.0176, Figure 6D,E), strengthening the correlation between OSAP induction and disease progression. 

Interestingly, in both series, the OSAP signature scores were statistically correlated with the Lgr5-GFP^high^ [29] and EphB2 [47] stem cell signatures, and with the score of a signature established on the basis of genes up-regulated in the stem-like subgroup of CRCs (stem-like score) [48], suggesting that the OSAP is activated in stem cells or progenitors in chronic inflammation (Figure 6C,F).

### 3.5. OSAP Is Induced in Intestinal Precursor Cells in Both Colitis-Associated and Sporadic Murine Colorectal Cancer Models

Several genes of the OSAP signature are also known to be active in immune cells, unveiling a potential interference of the inflammation index with OSAP scores. To circumvent this problem and assess whether OSAP induction takes place in the initiation of colorectal tumor development, we used the previously reported gene expression profiles of intestinal epithelial stem and precursor cells isolated from either normal colons or AOM/DSS-induced tumors isolated from *Lgr5-EGFP* mice exposed to azoxymethane/dextran sodium sulfate [28] (AOM/DSS, dataset GSE46200). The transcriptomic analysis was performed from cells which were flow-sorted into Lgr5^high^ and Lgr5^low^ fractions based on GFP expression. The Lgr5^high^ cell population includes stem cells and Lgr5^low^ are their immediate daughter cells (committed precursor cells), while differentiated intestinal cells are Lgr5^neg^. 

As we compared homogenous epithelial cell populations, the in vitro-established epithelial OSAP signature was used. Confident that the down-regulated genes resulted from the aberrant NF-κB activation in HCEC cells, the OSAP score was defined on the basis of its 27 up-regulated genes. The OSAP signature scores were significantly higher in adenoma-derived colonic stem/precursor cells than in their normal counterparts (Mann–Whitney test, *p* < 0.0001, Figure 7A), supporting the conclusion that the level of activation of the program increases during the early phases of tumor progression. In adenomas, the score was even higher in intestinal precursor cells, suggesting that the program is mainly induced in these cells.

We next extended our analysis to wild-type small intestine or *Apc*-mutant adenomas from Lgr5-EGFP-Ires-CreERT2/Apc^fl/fl^/R26R-Confetti mice [29] (dataset GSE37929). The OSAP signature scores were higher in small intestine adenoma-derived intestinal stem/early precursor cells (adenoma GFP^high^ and GFP^low^, respectively) than in normal stem cells (normal GFP^high^) (Mann–Whitney test, *p* < 0.0001, Figure 7B), suggesting a similar induction of the program in this experimental setting. Interestingly, in both series, the OSAP induction was concomitant to *Zeb1* and/or *Zeb2* expression (*Zeb1* Rho = 0.7944, *p* < 0.0001, and *Zeb2* Rho = 0.9235, *p* < 0.0001; *Zeb1* Rho = 0.7409 *p* < 0.0001, respectively), supporting the role of these embryonic transcription factors in orchestrating the program. 

## 4. Discussion

ROS, synthesized abundantly in inflammatory lesions, cause oxidative stress, leading to lipid, protein, and DNA base oxidation [49]. This repeated and continuous stress leads to telomere attrition and cell commitment to a senescence program [4,30]. The selection of p53 mutants in high-grade dysplasia is presumed to facilitate an escape from chronic-inflammation-induced senescence and thereby to promote tumor initiation [14]. We reasoned that, alternatively, a subset of intestinal stem cell/early progenitor cells could adapt to oxidative stress, and the resulting tolerance to oncogenic insults could facilitate their neoplastic transformation. In line with this hypothesis, a SIRT6-driven metabolic adaptation associated with dampened ROS was recently shown to facilitate the neoplastic transformation of a subset of intestinal stem cells [18]. By setting up an in vitro model, we first confirmed that submission of human colonic epithelial cells to oxidative stress invariably led to the emergence of resistant clones. The synchronized emergence of cells rather than individual clones is likely to reflect adaptation rather than the selection of preexisting cell subpopulations. Furthermore, depending on the experimental setting, the OSAP induction relies on either *ZEB1* or *ZEB2* induction. It is more likely that the two functionally related transcription-factor-encoding genes are induced in response to different stimuli, a context-dependent variability confirmed in vivo (Figure 7), and drive a similar genetic program rather than the selection of two distinct, preexisting populations, according to the stress-induced condition. 

Our observations argue in favor of the reversibility of senescence. Indeed, if senescence was originally considered to be an irreversible process, several emerging studies challenge this view, whether the senescence is telomeric [50,51], oncogene-induced [52] or therapy-induced [53,54,55]. As recently brilliantly reviewed, cellular senescence is an epigenetically remodeled and reversible stress response condition [56]. DNA-damaged or oxidative stressed cells undergo large-scale chromatin remodeling, involving histone 3 Lysine 9 trimethylation (H3K9m3) that stably represses S-phase-promoting genes and an upregulation of pro-inflammatory cytokines and chemokines that constitute the senescent-associated secretory phenotype (SASP). As demonstrated in mouse lymphoid cells, experimentally-induced depletion in H3K9 methyltransferase Suv39h1 is sufficient to revert therapy-induced senescence and reinitiate cell proliferation in vivo [53]. Interestingly, research on induced pluripotent cells demonstrate that senescence and stemness-reacquisition through genetic reprogramming are intimately linked by overlapping signaling networks [53,57]. Whether cell commitment to a senescence program is a prerequisite for OSAP induction is plausible, though it has thus far not been experimentally demonstrated. How the OSAP facilitates escape also requires further investigation. Obviously, several proteins, including QPRT and APOE, contribute to it. Indeed, with their ability to fuel NAD and dampen NF-κB activity, respectively, these two proteins were reported to contribute to resistance to oxidative stress [43,58]. Of note, among the NF-κB targets downregulated in reprogrammed HCEC cells, several encode components of the SASP (*CCL20*, *CXCL1*, and *CXCL2,* encoding for MIP-3a, GROα, and GROβ, respectively), known to be essential to maintaining cell commitment in a senescence program [59,60].

We next addressed the significance of this reprogramming in IBD patients by ssGSEA. We confirmed that the program was induced in inflamed mucosae of IBD patients and correlated with previously reported stem-cell signatures [29,47,48], suggesting that this cell reprogramming takes place in either intestinal stem or precursor cells. By scoring the OSAP in intestinal epithelial cells that were flow-sorted from chronic-inflammation and sporadic colorectal carcinogenesis mouse models on the basis of lineage-tracing experiments, we confirmed that the score was higher in intestinal progenitor cells isolated from adenomas compared to their normal counterparts in both murine carcinogenesis models [28,29]. Although a signature score depends on the algorithm used and the state of the cells and can only be compared between samples from the same analysis, the positive scores observed in adenomas from AOM/DSS-treated mice supports a preponderant role for inflammation in the activation of OSAP.

Interestingly, ZEB1 was shown to be induced in chronically inflamed intestinal mucosa and to promote intestinal inflammation and colitis-associated colorectal cancers through the repression of the N-methyl-purine glycosylase MPG-encoding gene [61]. This enzyme is involved in the recognition and excision of DNA damage, and its loss of function is associated with higher susceptibility to DNA damage, inflammation, and tumor development [62]. Moreover, *ZEB1* expression in inflamed or malignant epithelial cells induces higher ROS and inflammatory cytokine production by mucosal immune cells, strengthening a positive feedback loop between the two cellular compartments [61]. We propose that increased tolerance to ROS through OSAP induction also contributes to the ZEB1 oncogenic arsenal [63]. As orchestrators of genetic programs, ZEB proteins differentially promote tumor initiation according to the cellular context. In melanocytes, a shift from ZEB2 to ZEB1 was previously shown to modulate the balance between cell differentiation and proliferation and to thereby drive tumor initiation and secondary site colonization [64,65]. In human mammary epithelial cells (HMEC-hTERT), both ZEB proteins similarly promote cell commitment to EMT, favoring their neoplastic transformation [19]. The determinants between the OSAP and EMT program-induction in mammary and colonic epithelial cells are yet to be determined. Further investigations are needed to decipher the underlying mechanisms.

The etiopathology of inflammatory bowel diseases is multifactorial, involving genetic predisposition, mucosal barrier dysfunction, and an alteration of the microbiota composition (dysbiosis) and of the immune system as well as environmental and lifestyle factors [66]. Multiple approaches have been developed to attempt to resolve inflammation, including pharmacological treatments with aminosalicylates and oral corticoids and the use of immunomodulators, pro-inflammatory cytokine inhibitors (anti-TNF, and anti IL22-23 therapies) or small molecules (e.g., JACK inhibitors). Novel therapies improving intestinal microecology with antibiotics, probiotics, prebiotics, postbiotics, synbiotics, and fecal microbiota transplantation have additionally emerged [67]. Unfortunately, the subclinical, persistent inflammation leads to conventional and non-conventional dysplasia and an increased risk of cancer development. The identification of intestinal cell reprogramming as an adaptation process to reactive oxygen species in the time course of their malignant conversion offers the perspective that several of the associated proteins could, in the future, be used as early predictive markers of a risk of developing cancer.

## 5. Conclusions

We herein demonstrated that intestinal epithelial cells can adapt to chronic inflammation through a partial genetic reprogramming orchestrated by the ZEB transcription factor in vitro. We confirmed the biological relevance of this reprogramming in Crohn’s and ulcerative colitis disease patients by ssGSEA and demonstrated that the program is activated in intestinal precursor cells in the early steps of murine intestinal carcinogenesis, especially during chronic inflammation. These data sustain the determinant role of cell plasticity in the ability of intestinal stem/early precursor cells to adapt to stress, and is likely what Sphyris and collaborators named their emancipation [68], gain of autonomy to the niche during tumor initiation.

## Figures and Tables

**Figure 1 cancers-15-00973-f001:**
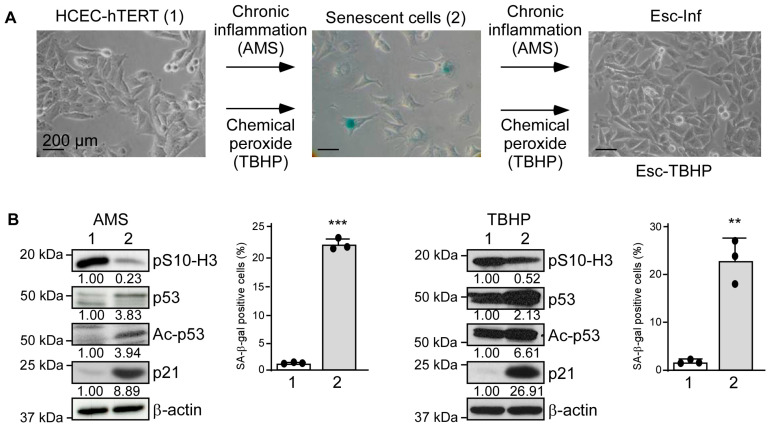
Human colonic epithelial cells adapt to chronic inflammation-associated oxidative stress in vitro. (**A**) HCEC-hTERT cells (left panel) were subjected to chronic inflammation by culturing them in the presence of activated macrophage supernatant (+AMS) or tert-Butyl-hydroperoxide (TBHP). Cells committed to senescence after a week, demonstrated by the detection of SA-β-galactosidase activity (central panel), and maintained their senescent phenotype for three weeks before resuming proliferation (right panel). (**B**) Assessment in both experimental settings (left AMS, right TBHP) of phospho-serine 10 histone H3 (pS10-H3), p53, acetylated-p53 (Ac-p53), and p21^CIP1^ (p21) by Western blot (protein levels were quantified relative to the β-actin and HCEC-hTERT cells) and of SA-β-galactosidase-positive cells in parental HCEC-hTERT cells (1) and senescent cells (2). A non-parametric Student’s *t*-test, mean, and SD of one experiment performed in triplicate are shown. *** *p* < 0.001, ** *p* < 0.01. Uncropped Western blots are provided in Appendix A.

**Figure 2 cancers-15-00973-f002:**
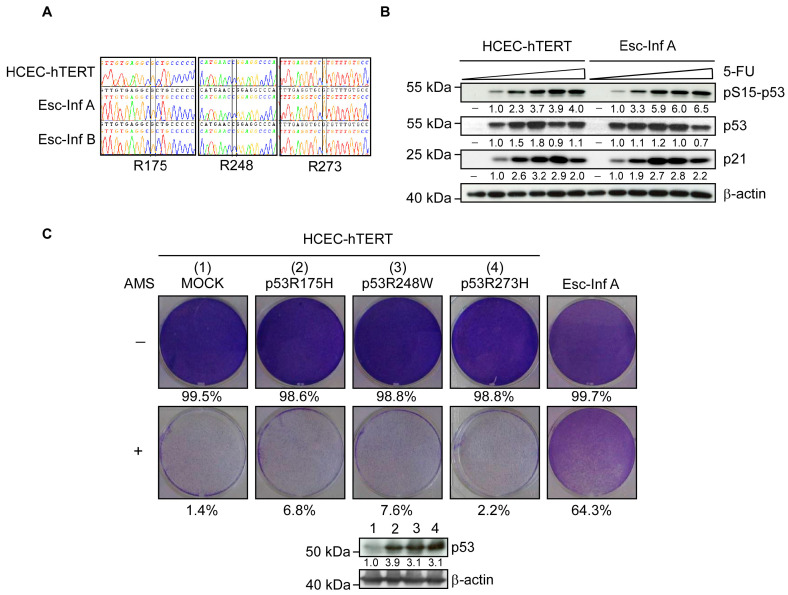
Escape from chronic-inflammation-induced senescence does not rely on p53 inactivation. (**A**) Confirmation of the wild-type status of the *TP53* gene in HCEC-hTERT cells and their derivatives. *TP53* exons within parental HCEC-hTERT cells and their Esc-Inf derivatives (of two independent experiments, Esc-Inf A and B) were fully sequenced. No mutation in *TP53* was found, as shown by the sequences encompassing the exons encoding the residues R175, R248, and R273 that constitute the main mutation hot-stops reported in colitis-associated colon cancers. (**B**) The functionality of the p53 pathway in both parental HCEC-hTERT and its Esc-Inf derivatives was compared by submitting cells to an increasing dose of Fluorouracil (5-FU) over 24 h. The phosphorylation of the Ser10 of p53 (pS15-p53) and the levels of p53 and p21^CIP1^ (p21) were assessed by Western blot. Protein levels were quantified relative to the β-actin and HCEC-hTERT cells. (**C**) The ectopic expression of p53 variants does not facilitate the escape of HCEC-hTERT cells from chronic-inflammation-induced senescence. Upper panels: cells were successively infected with mutant p53 retroviral-expressing vectors (as indicated on top), cultured in the presence of activated macrophage supernatants (+AMS) for ten days, and stained with crystal violet. Esc-Inf cells were used as an internal positive control. Percentages of stained surface are indicated. Lower panel: analysis of p53 by Western blot. Protein levels were quantified relative to the β-actin and HCEC-hTERT cells. Uncropped Western blots are provided in Appendix A.

**Figure 3 cancers-15-00973-f003:**
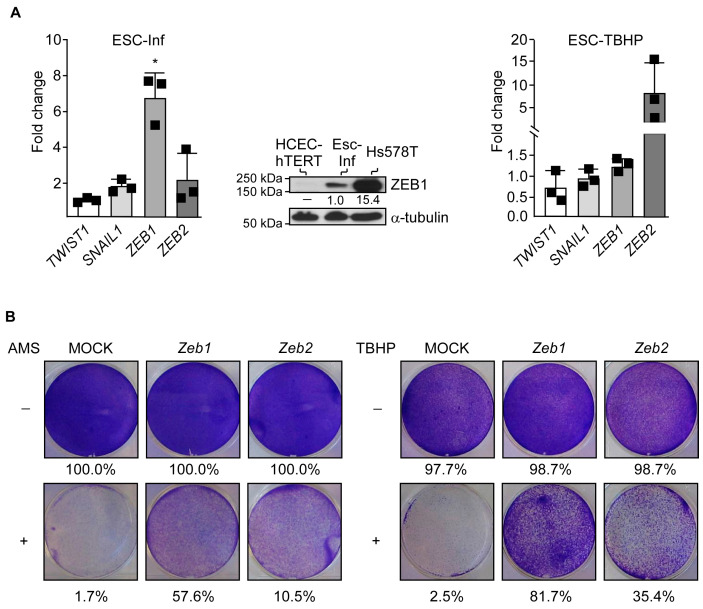
Escape from chronic-inflammation-induced or TBHP-induced senescence relies on the induction of *ZEB* genes. (**A**) Relative expression of *SNAI1*, *TWIST2*, *ZEB1,* and *ZEB2* in cells that escaped from chronic-inflammation- (left panel) or TBHP-induced senescence (right panel), as assessed by qRT-PCR. Levels are expressed relative to the housekeeping *HPRT1* gene and are normalized against HCEC-hTERT cells. One sample t and Wilcoxon test and the mean and SD of three independent experiments are shown. * *p* < 0.05 Central panel: analysis of ZEB1 in Esc-Inf cells by Western blot. EMT-committed Hs578T breast cancer cells were used as a positive protein detection control. Protein levels were quantified relative to the α-tubulin and Esc-Inf cells. (**B**) Ectopic expression of murine *Zeb1* or *Zeb2* facilitates escape from chronic inflammation-induced (+AMS, left panels) or chemical, peroxide-induced (+TBHP, right panels) senescence, as assessed in a crystal violet assay. Percentages of stained surface are indicated. Uncropped Western blots are provided in Appendix A.

**Figure 4 cancers-15-00973-f004:**
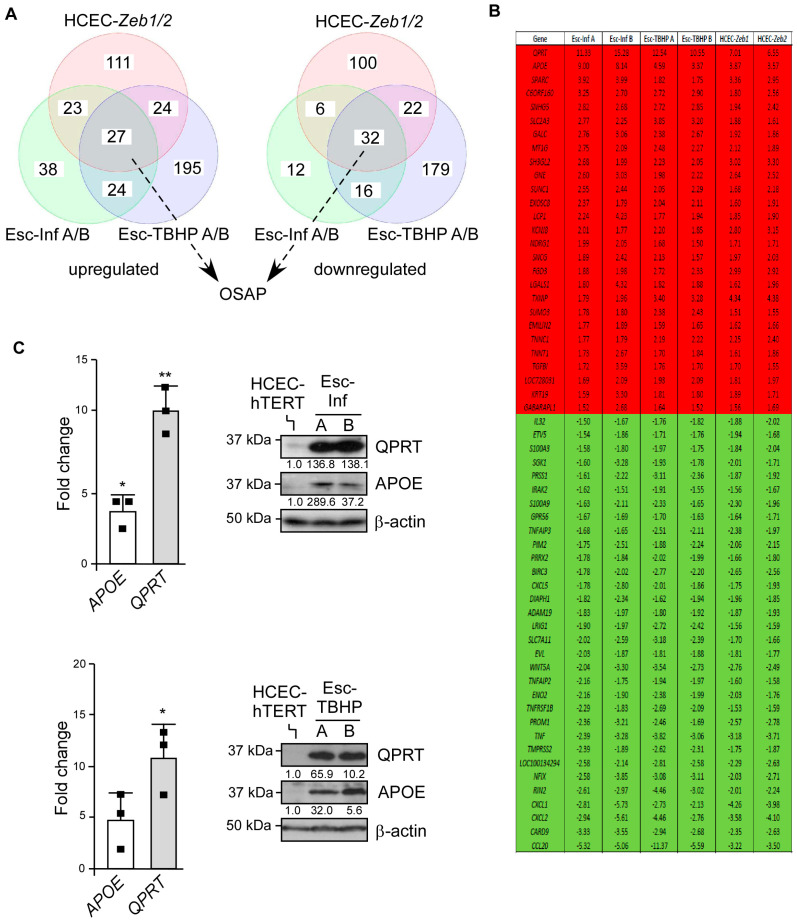
The ZEB-driven resistance relies on a partial genetic reprogramming. (**A**) Venn diagram highlighting the number of genes commonly up- or down-regulated (cut-off > 1.5) in HCEC-hTERT cells which had either escaped from chronic-inflammation-induced senescence (Esc-Inf A or B from two independent experiments), TBHP-induced senescence (Esc-TBHP A or B from two independent experiments), or ectopically expressed either *Zeb1* or *Zeb2* (HCEC-*Zeb*1 and HCEC-*Zeb*2) when compared to their HCEC-hTERT parental counterparts. Common up- and down-regulated genes constitute the basis for the OSAP program. (**B**) Listing of the genes commonly up-regulated (in red) or down-regulated (in green) in Esc-Inf (A and B), Esc-TBHP (A and B), HCEC-*Zeb1*, and HCEC-*Zeb2* compared to the parental HCEC-hTERT cells (cut-off ≥ 1.5). (**C**) Expression analysis of *APOE* and *QPRT* in cells that escaped either from chronic-inflammation-induced (upper panels) or TBHP-induced (lower panels) senescence. Left panels: expression analysis by qRT-PCR. Levels are expressed relative to the housekeeping *HPRT1* gene and are normalized against HCEC-hTERT cells. One sample t and Wilcoxon test and the mean and SD of three independent experiments are shown. Right panels: analysis of APOE and QPRT by Western blot. Protein levels were quantified relative to the β-actin and HCEC-hTERT cells. ** *p* < 0.01, * *p* < 0.05. Uncropped Western blots are provided in Appendix A.

**Figure 5 cancers-15-00973-f005:**
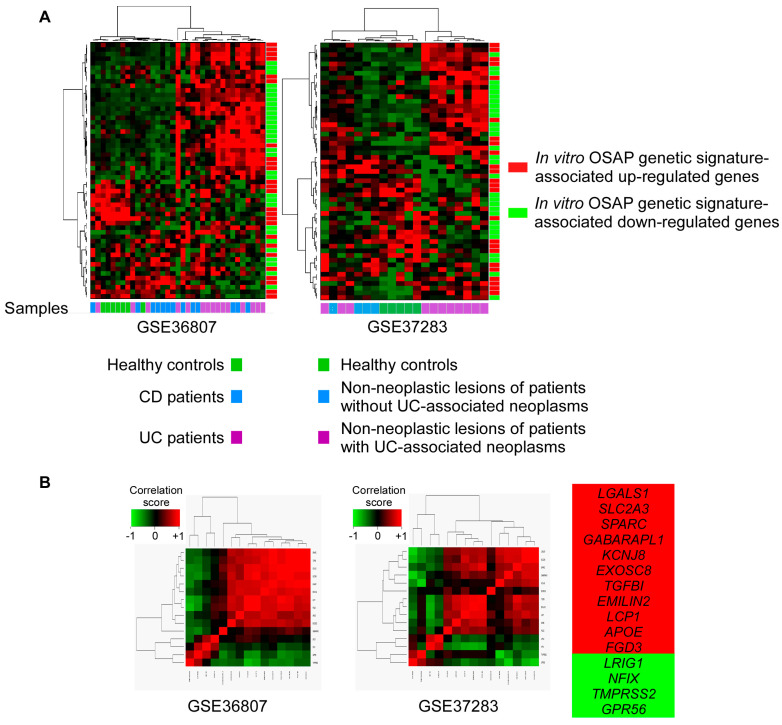
Analysis of OSAP in UC and CD mucosa. (**A**) Unsupervised hierarchical clustering of healthy controls, UC, and CD mucosa (datasets GSE36807 and GSE37283) using the in vitro OSAP signature. (**B**) Among the in vitro established OSAP-signature-associated genes, eleven and four genes (as listed on the right) are overexpressed (estimate [UC-control] > 0, labeled in red) and underexpressed (estimate [UC-control] < 0, labeled in green) in UC mucosa versus healthy mucosa in the dataset GSE36807 and in UC mucosa with associated neoplasms versus healthy mucosa in the dataset GSE37283, as assessed by a Wilcoxon test.

**Figure 6 cancers-15-00973-f006:**
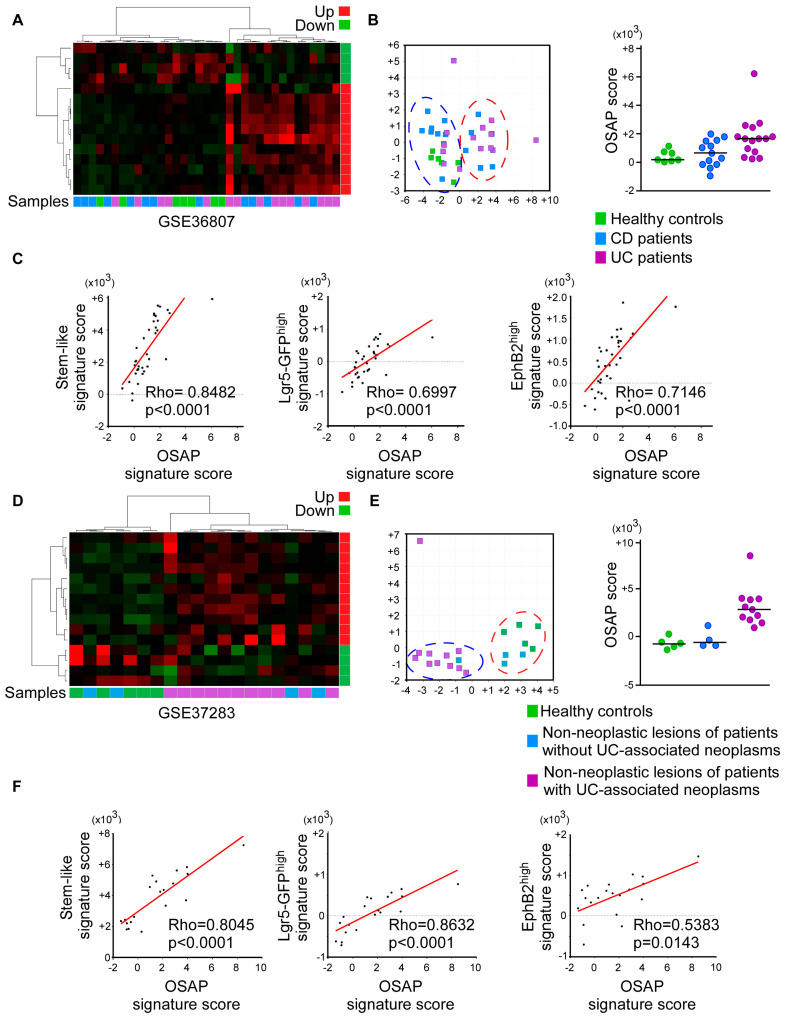
OSAP is recurrently activated in ulcerative colitis and Crohn’s disease patients. Analysis of the biological significance of the OSAP signature in UC and CD cases (datasets GSE36807 and GSE37283, as indicated). (**A**,**D**) Hierarchical clustering analysis using OSAP genes showing the differential relative expression of the OSAP signature-associated genes in patient samples. (**B**,**E**) Left panels: distribution of samples using principal component analysis. Discriminating genes were used to generate a two-dimensional plot of the data. Right panels: enrichment scores of the OSAP signature were computed in all samples from the three groups using ssGSEA and then compared using a Kruskal–Wallis test (*p* < 0.0001). (**C**,**F**) Spearman correlation between the OSAP signature scores and stem-like, Lgr5-GFP^high^, or EphB2^high^ signature scores. Coefficients of correlation and *p*-values are shown.

**Figure 7 cancers-15-00973-f007:**
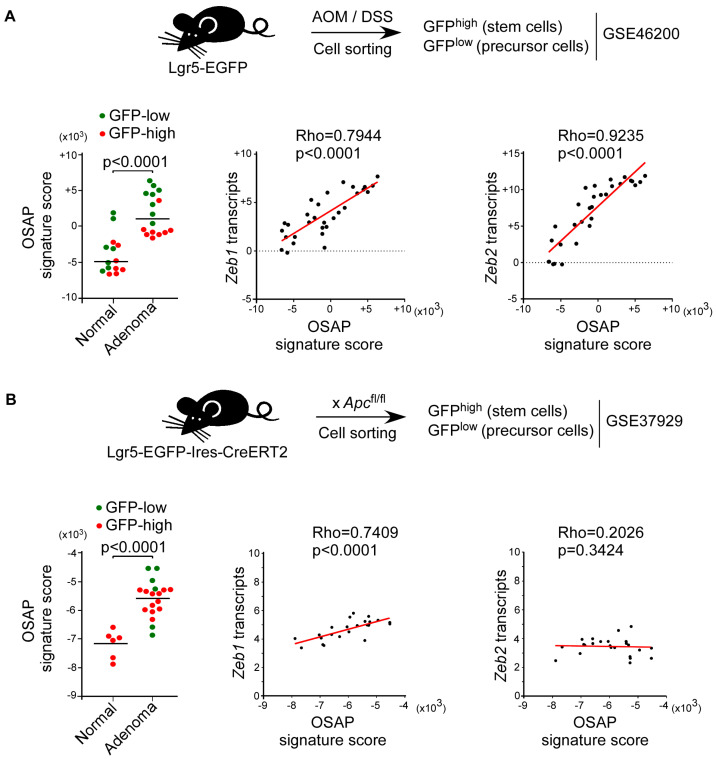
OSAP is induced in murine intestinal stem/early precursor cells in response to a chronic inflammation or to the loss of *Apc*. Analysis of OSAP in intestinal stem and precursor cells sorted from AOM/DSS treated-derived adenoma or control littermates (dataset GSE46200) (**A**) or sorted from *Apc*-deficient, mouse-derived adenoma or control littermates (dataset GSE37929) (**B**). Left panels: OSAP signature scores in intestinal stem cells (GFP^high^) and precursors (GFP^low^) sorted from either adenoma or control littermates (normal). Mann–Whitney test is shown. Center and right panels: Spearman correlation between *Zeb1 or Zeb2* expression and OSAP signature scores. Coefficients of correlation and *p*-values are shown.

## Data Availability

The complete set of raw and normalized files of cell line transcriptomes is available at the GEO database under accession number GSE70170.

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
