# Peer review of "Intestinal Epithelial Cells Adapt to Chronic Inflammation through Partial Genetic Reprogramming"

_cancers, 2023, doi:10.3390/cancers15030973_

Round 1

Reviewer 1 Report

The reviewed manuscript is interested to read, presents robust hypothesis, sound results and relevant conclusions. It is a complete and nice work. Just one short comment-question, authors present in there in vitro experiments results - colonic cells induced into stress-related senescent phenotype which despite the presence of a stressor could return to proliferation cycle. Is it meant that particular stress conditions may stop some living systems (cells) in quasi-senescence resembling GO phase which is fully reversible despite continued presence of a causing agent? Often, stress-induced senescent phenotype is passed for an irreversible event akin to classical replicative senescence. Data shwown here may argue against this suggestion and perhaps deserve bolder discussion.

The text needs to be proofread for truly minor issues.

Author Response

The reviewed manuscript is interested to read, presents robust hypothesis, sound results and relevant conclusions. It is a complete and nice work.

We thank the reviewer for his/her appreciation.

Authors present in there in vitro experiments results-colonic cells induced into stress-related senescent phenotype which despite the presence of a stressor could return to proliferation cycle. Is it meant that particular stress conditions may stop some living systems (cells) in quasi-senescence resembling GO phase which is fully reversible despite continued presence of a causing agent? Often, stress-induced senescence is passed to be irreversible event akin to classical replicative senescence. Data shown here may argue against this suggestion and perhaps deserve bolder discussion.

The stress conditions imposed to HCEC cells are sublethal. Indeed, higher dosage of macrophage supernatant or TBHP induces massive apoptosis. Culture of cells in sublethal condition is commonly used to enforce them to adapt and identify escape mechanisms. To clarify this point, we have included in the result section the following sentence: “Conditions were set up to submit cells to sublethal doses, experimental conditions commonly used to enforce cell adaptation and unveil escape mechanisms”.

We agree that the point of the reversibility of senescence deserves to be addressed in the discussion section.

We have now included in the discussion the following text:

“Our observations argue in favor of the reversibility of senescence. Indeed, if senescence was originally considered as an irreversible process, several emerging studies challenge this view, whether it is telomeric[50,51], oncogene-induced[52] or therapy-induced senescence[53–55]. As recently brilliantly reviewed, cellular senescence is an epigenetically remodeled and reversible stress response condition[56]. DNA-damaged or oxidative stressed cells undergo large-scale chromatin remodeling involving histone 3 Lysine 9 trimethylation (H3K9m3) that stably repress S-phase-promoting genes and an upregulation of pro-inflammatory cytokines and chemokines that constitute the senescent-associated secretory phenotype (SASP). As demonstrated in mouse lymphoid cells, experimentally-induced depletion in H3K9 methyltransferase Suv39h1 is sufficient to revert therapy-induced senescence and reinitiate cell proliferation in vivo[53]. Interestingly, research on induced pluripotent cells demonstrates that senescence and stemness-reacquisition through genetic reprogramming are intimately linked by overlapping signaling networks[53,57]. Whether cell commitment to a senescence program is a prerequisite for OSAP induction is plausible although it has so far not been experimentally demonstrated. How OSAP facilitates escape also needs further investigation. Obviously, several proteins, including QPRT and APOE contribute to it. Indeed, their ability to fuel NAD and dampen NF-κB activity, respectively, these two proteins were reported to contribute to resistance to oxidative stress [43,58]. Of note, among the NF-κB targets downregulated in reprogrammed HCEC cells, several encode components of the SASP (CCL20, CXCL1, CXCL2 encoding respectively for MIP-3a, GROα, and GROβ), known to be essential to maintain cell commitment in a senescence program[59,60].”

The text needs to be proofread for truly minor issues

We apologize for that and have ensured that typos have been corrected.

Reviewer 2 Report

The manuscript is well organized and the conclusions are coherent.

I suggest to the authors to prepare a graphical abstract, in this way that the results are immediate and therefore can better help the reader of the manuscript.

The authors perhaps should read these recent manuscripts because diet can also influence the microbiota.
Ye?ilyurt N, Y?lmaz B, A?agündüz D, Capasso R, Microbiome-based personalized nutrition as a result of the 4.0 technological revolution: A mini literature review. Process Biochemistry, 2022 121, 257-262

A?agündüz D, Cocozza E, Cemali Ö, Bayaz?t AD, Nanì MF, Cerqua I, Morgillo F, Sayg?l? SK, Berni Canani R, Amero P and Capasso R (2023), Understanding the role of the gut microbiome in gastrointestinal cancer: A review. Front. Pharmacol. 14:1130562

The manuscript would benefit from inclusion of introducing/bridging sentences between the individual parts of the "Results" that explain the logical order and rationale for the experiments

The authors write :"To assess whether OSAP induction takes place in the initiation of colorectal tumor development, we
used the previously reported gene expression profiles of intestinal epithelial stem and precursor cells isolated from either normal colons or AOM/DSS-induced tumors isolated from Lgr5-EGFP mice exposed to azoxymethane/dextran sodium sulfate[28]" if I understand correctly, the authors , they didn't repeat the experiments, the data dates back to 2014. Can they clarify this aspect?Perhaps they should have done the treatment and then isolated the cells.

In the Discussion, the Authors should highlight the possible clinical significance of their findings

Author Response

The manuscript is well organized and the conclusions are coherent.

We thank the reviewer for his/her appreciation.

I suggest to the authors to prepare a graphical abstract, in this way that the results are immediate and therefore can better help the reader of the manuscript.

We thank the reviewer for this suggestion. We have modified the graphical abstract to make it as explanatory as possible.

The authors perhaps should read these recent manuscripts because diet can also influence the microbiota.

In the discussion the authors should highlight the possible clinical significance of their findings

The manuscript is specifically focused on the consequences of reactive oxygen species associated with chronic inflammation. We agree that we did not address the different strategies currently developed to resorb chronic inflammation in inflammatory bowel disease patients and discuss the clinical significance of our results.

Although the two mentioned references are of quality, numerous reviews address different aspects of the link between microbiota, IBD and colorectal cancers. To limit the number of references, we focused on a single reference, the recent review by Cai Z. et al., 2021, summarizing all therapeutic strategies currently used.

We propose to add the following text at the end of the discussion section.

“The etiopathology of inflammatory bowel diseases is multifactorial, involving genetic predisposition, mucosal barrier dysfunction, alteration of the microbiota composition (dysbiosis) and of the immune system as well as environmental and lifestyle factors[66]. Multiple approaches have been developed to attempt to resolve inflammation including pharmacological treatments with aminosalicylates and oral corticoids, use of immunomodulators, pro-inflammatory cytokine inhibitors (anti-TNF, anti IL22-23 therapies) or small molecules (e.g. JACK inhibitors). Novel therapies improving intestinal microecology with antibiotics, probiotics, prebiotics, postbiotics, synbiotics and fecal microbiota transplantation have additionally emerged[67]. Unfortunately, the subclinical persistent inflammation leads to conventional and non-conventional dysplasia with an increased risk of cancer development. The identification of intestinal cell reprogramming as an adaptation process to reactive oxygen species in the time course of their malignant conversion, offers the perspective that several of the associated proteins could in the future be used as early predictive markers of a risk of developing cancers.”

The manuscript would benefit from inclusion of introducing/bridging sentences between the individual parts of the results that explain the logical order and rationale for the experiments.

We thank the reviewer for this constructive remark. Such sentences have now been included.

The authors write: “To assess whether OSAP induction takes place in the initiation of colorectal tumor development, we used the previously reported gene expression profiles of intestinal epithelial stem and precursor cells isolated from either normal colons or AOM/DSS-induced tumors isolated from Lgr5-EGFP mice exposed to azoxymethane/dextran sodium sulfate”. If I understand correctly, the authors, they didn’t repeat the experiments, the datadates back to 2014. Can they clarify this aspect? Perhaps they should have done the treatment and then isolated the cells.

The study of Daniela Hirsch and colleagues (Hirsch et al., 2014) has been cited 115 times (Web of Science, 01/2023). The quality of their cell sorting was confirmed by their gene expression profiles, distribution of stem cell markers such as Lgr5, Smo2, EphB2, and the differential WNT pathway activation between stem and progenitor cells (Hirsch et al., 2014). We decided to use these analyses instead of reproducing the experiments for ethical (according to the “Replacement, Reduction, Refinement” rule), feasibility (the use of AOM in our animal facility is forbidden for security reason) and financial reasons.
